A scheme of hiding large-size image into small-size image based on FCdDNet

Liu Lianshan liulianshan@sdust.edu.cn
Tang Li
http://orcid.org/0009-0006-3300-7542 Tong Shanshan
Huang Yu
College of Computer Science and Engineering, Shandong University of Science and Technology , Qingdao, Shandong , China
Dong Shi
Electronic publication date: 2024 Jun 25
Publication date: 2024
Volume: 10
Electronic Location ID: e2140
Received 2024 Jan 2; Accepted 2024 May 28
Copyright: © 2024 Liu et al.
Copyright year: 2024
Copyright holder: Liu et al.
License: This is an open access article distributed under the terms of the Creative Commons Attribution License, which permits unrestricted use, distribution, reproduction and adaptation in any medium and for any purpose provided that it is properly attributed. For attribution, the original author(s), title, publication source (PeerJ Computer Science) and either DOI or URL of the article must be cited.
License URL: https://creativecommons.org/licenses/by/4.0/

Keywords: Information hiding, Deep learning, Large capacity, FCdDNet

Funding: Shandong Natural Science Foundation ZR2022MF277 Shandong Province Key R&D Plan (Soft Science) 2023RKL01003 This work was supported by the Shandong Natural Science Foundation (No. ZR2022MF277), and the Shandong Province Key R&D Plan (Soft Science) Project (No. 2023RKL01003). The funders had no role in study design, data collection and analysis, decision to publish, or preparation of the manuscript.

==============================
The hiding capacity of the current information hiding field has reached a relatively high level, which can hide two color images into one color image. In order to explore a larger hidden capacity, an information hiding scheme based on an improved FCdDNet is proposed, which can hide large-size color images into small-size color images. An improved FCdDNet network is used as the main structure shared by the hidden network and the extraction network. These two networks promote and improve each other during the confrontation training process and are used in pairs. It can be seen that the proposed scheme achieves a larger information hiding capacity, and the hidden information is four times larger than the scale of the carrier image. At the same time, the visual effect after hiding is guaranteed, and the image extracted from the hidden image also has a high degree of restoration. The scheme can be applied to image authentication, secret image transmission, and other fields.

Introduction

Today, due to the popularization of network informatization, the security of Internet communications has become the focus of attention. Image information hiding can conceal classified data or images in public images, and it is difficult for people to intuitively perceive the existence of such secrets. It is important to solve this information security problem. Current information concealment technology is used in many fields, such as secret information transmission, copyright identification, medical image archiving, and communication (Wang et al., 2017), etc.

Traditional information hiding technology made use of the correlation between pixels, and some schemes directly manipulated the image, such as LSB (Shehzad & Dag, 2019), histogram correction (Fallahpour, 2007), and other schemes. Some schemes first performed a series of transformations on the image and then operated on the transformation coefficients (Abdel-Aziz, Hosny & Lashin, 2021). Common transformations include wavelet transformation (Meng et al., 2021), discrete cosine transformation, etc. In order to improve security, there are also some schemes that use encryption schemes in information hiding, which were divided into the schemes of Reserving Room Before Encryption (Shiu et al., 2019) and Reserving Room After Encryption (Dragoi, Coanda & Coltuc, 2017).

With the rapid development of deep neural network (DNN) technology, a new method of using DNN for information hiding has been proposed (Brandao & Jorge, 2016). Rehman et al. (2018) designed an end-to-end trained encoder-decoder network for image steganography and transformed hidden grayscale images into color images of the same size. Current information hiding schemes are mostly designed based on the encoder-decoder concept. Chen et al. (2020) and Yang et al. (2020) used the generative adversarial network (GAN) network for information hiding. The GAN network was originally designed for image generation, so it can have a higher visual effect when generating hidden images. Wang et al. (2019) used the GAN network to propose HidingGAN for information hiding. The proposed Inception-ResNet block could better integrate the secret image into the carrier image so that the obtained secret image will be more confidential.

Due to the application of DNN in the field of information hiding, the hidden capacity has been greatly improved. Nowadays, it is possible to hide grayscale images into color images (Li et al., 2020), and even hide color images into color images of the same size (Duan et al., 2020, 2021) which could still have a good visual effect. Baluja (2017) proposed the scheme of Hiding Images in Plain Sight, which realized the hiding of images of the same size. He believed that similar schemes are to realize hiding by compressing secret information and distributing it into the carrier image. In subsequent research Baluja (2020), expanded the application of this scheme, not only to keep the hidden information safe but also to hide multiple images.

The U-Net scheme proposed by Ronneberger, Fischer & Brox (2015) solved the problems of vanishing and exploding gradients. The U-Net network had a unique cross-layer connection method, in which the image characteristics of the underlying network are supplemented and the characteristics can be reused. U-Net has a good segmentation effect in the field of medical image segmentation, and its idea of cross-layer connection has been widely recognized and used in the field of segmentation. Pandey & Wang (2020) proposed a fully convolutional neural network with dense connections and dilated convolutions, abbreviated as FCdDNet in this article, which combined the excellent ideas of U-Net and other programs. Compared with the traditional U-Net, SegNet, FC-DenseNets, and other schemes, it had a great improvement. In addition, the solution had a small network scale and and can realize real-time speech enhancement. Duan et al. (2019) used the U-Net model to hide color images into color images. This scheme used the U-Net cross-layer connection method in the hiding network, and there is basically no visual difference between the obtained hidden image and the original carrier image. Subramanian et al. (2021) proposed a lightweight deep convolutional autoencoder architecture to embed a secret image into the carrier image and extract the embedded secret image from the hidden image, which also achieved high hiding ability, security, and robustness and imperceptibility.

Huang et al. (2017) proposed the DenseNet network model in 2017, which solved the problem of gradient explosion caused by the superposition of layers. DenseNet dense connection all the front layers with the back layers, realized feature reuse. This scheme had better effect and performance than ResNet proposed by He et al. (2016). Liu et al. (2020) used the DenseNet network to implement an image steganography scheme, which had better visual effects and robustness, but the hidden capacity was not much improved compared to the traditional scheme. Huo et al. (2024) proposed a chaotic map-enhanced image steganography network (CHASE), which first hides the color image in the grayscale image and reduces the difference between the container image and the carrier image through image replacement, thereby enhancing the capacity and security of steganography. Guan et al. (2022) proposed a inversible multi-image hidden neural network (DeepMIH), which innovatively models image hiding and extraction as the forward and backward processes of the inversible neural network, and also designed an importance graph module to guide the hiding of multiple images, which has greatly improved the hiding capacity compared with traditional schemes.

In the field of information hiding, hidden capacity has always been a major research hotspot. To further explore the capacity of hidden images, an information hiding scheme based on improved fully convolutional dense dilated network (FCdDNet) is proposed in this article. The improved FCdDNet network is the main structure shared by information hiding and extraction. During the hiding process, the features of the large-size secret image are extracted and compressed multiple times and are then hidden in the small-size carrier image to obtain a hidden image. During the extraction process, the secret features contained in the hidden image are in turn extracted and decompressed to obtain a large-size extracted image. Therefore, compared with the current steganography methods, the proposed scheme can hide the secret data that is 4 times larger than the carrier image, and the visual effect is better than some small-volume schemes, and the information extraction accuracy of the proposed algorithm is also improved compared with the existing small-volume schemes.

Related works

DenseNet

Huang et al. (2017) proposed a DenseNet network model, in which the dense block used multiple internal connections to make full use of image features. This solution solved the gradient explosion problem caused by the number of overlay network layers at that time and also reduced a lot of parameters. The 5-layer dense block model with a growth rate of 4 proposed by Huang et al. (2017) is shown in Fig. 1.

Figure 1 The 5-layer dense block model.

The connection method in the dense block model is a channel-level connection operation, which is different from the shortcut exhibited in ResNet. This connection method features concatenation instead of an element-level addition operation. The input of each layer of the dense block network model comes from the output of all previous layers, which fully guarantees the information transmission between layers and maximizes the reuse of features. Compared with ResNet and other solutions, DenseNet has higher efficiency in solving the problem of gradient explosion. It has become the current mainstream method and is being used in various application scenarios of DNN.

FCdDNet

Pandey & Wang (2020) proposed a fully convolutional dense dilated network (FCdDNet) for real-time speech enhancement in the time domain. The network combines dense connectivity and dilated convolution. It also shows a dense dilated block and optimizes the loss function. The network has a smaller network scale and higher efficiency as compared to mainstream solutions such as U-Net and FC-DenseNets while ensuring the objective intelligibility and quality scores of model. The structure of FCdDNet is shown in Fig. 2.

Figure 2 The structure of FCdDNet.

The network architecture is an encoder-decoder based architecture with skip connections. Densely connected blocks with dilated convolutions are added after each layer of the encoder and decoder, and the dilated and densely connected blocks facilitate remote context aggregation over signals of different resolutions. The structure of the proposed dense dilated block is shown in Fig. 3.

Figure 3 The proposed dense dilated block.

The proposed dense dilated block combines the DenseNet network and the dilated convolutional network, in which the dilation rate of dilated convolution increases with the number of layers. Dense dilated block utilizes DenseNet’s feature reuse and gradient avoidance guarantee. In addition, expanded convolution is used to expand the receptive field of the convolution kernel while keeping the number of parameters unchanged, so as to avoid losing a large amount of local information.

The network uses the cross-layer connection method like U-Net many times to connect to the dense dilated block and other layers. Meanwhile, the cross-layer connection can also play a supplemental role in the underlying features and help to fill in the information loss generated in the feature extraction process. According to experimental comparison and analysis, FCdDNet possesses accuracy comparable to the most advanced networks such as FC-DenseNets, but the network scale and parameters are significantly reduced. Therefore, this scheme can maintain excellent efficiency while maintaining high accuracy in the field of medical image segmentation.

Method

The proposed scheme adopts the improved FCdDNet as the main structure, and combines more advanced networks, such as DenseNet and concatenate connections, instead of simple sequential connections, which can theoretically have a better feature reuse rate and thus ensure the accuracy of information extraction. In addition, the proposed hidden network adds a secret information compression layer before the main structure of the improved FCdDNet, compresses the original secret image to 1/4 of the original size and then performs the information hiding operation, so as to realize large-capacity information steganography. In the universal approximation theorem of neural networks, for a given continuous function or mapping, a multilayer feedforward network containing enough neurons can be used to approximate it to any accuracy, so that steganography networks can make the stego infinitely close to the carrier image, thus achieving high-quality image steganography.

Network structure

This article proposes an improved FCdDNet network for information hiding and information extraction. An improved FCdDNet network is adopted as the common main structure of the hidden and extraction networks. Compared with the original FCdDNet, in the overall structure, the improved FCdDNet reduced three layers of convolution or deconvolution operations and 2 cross-layer concatenations. The structure of the hiding network and extraction network formed by improved FCdDNet are shown in Fig. 4, and the network parameters are shown in Table 1.

Figure 4 Hidden network and extraction network structure formed by improved FCdDNet (specific parameters are shown in Table 1).

Table 1 The parameters of hiding and extraction network.

Layers	OutSize & OutFeaturesa	Type	
	Hiding	Extraction		
0	256 × 256, 6	–	Transition	
1	256 × 256, 32b	Convolution	
2	256 × 256, 16	Convolution	
3	128 × 128, 48	Transition	
4	128 × 128, 16	Dense dilated block	
5	64 × 64, 64	Transition	
6	64 × 64, 32	Convolution	
7	128 × 128, 16	Deconvolution	
8	128 × 128, 40	Convolution	
9	256 × 256, 20	Deconvolution	
10	256 × 256, 34	Convolution	
11	256 × 256, 3	–	Convolution (1 × 1)	
12	–	512 × 512, 3	Deconvolution	
Notes:

a OutSize & OutFeatures represent the size of the output feature and the number of layers of the output feature.

b Layers 1–10 share the same structure as the hidding and the extraction network, and have the same parameters.

The first improvement lies in the input and output process. Since the information hiding scheme involved two network structures, applying the same subject can ensure network balance during training. Among them, Layer 1 to Layer 10 are the main structures that hide and extract the network, while Layer 0 and Layer 11 are unique to the hidden network. Layer 12 is unique to the extraction network. Except for Layer 11 and Layer 12, all other layers in the scheme use batch normalization (BN) to make the results of each layer more evenly distributed. All convolution operations used ReLU for activation, and all deconvolution operations used LeakyReLU for activation.

Secondly, we abandon the asymmetric architectures and used 3 × 3 convolutions uniformly. The asymmetric architecture can reduce the computational complexity for feature maps of medium network size (between 12 and 20), so it is not suitable for this scheme. The convolution layer used in this article includes a convolution operation with a 3 × 3 convolution kernel, BN acceleration, and ReLU activation. The deconvolution layer consists of a deconvolution operation with a convolution kernel of 3 × 3, BN acceleration, and LeakyReLU activation.

Then, in order to adapt to the high precision of the information hiding scheme and to avoid information loss, the Transition layer is improved to remove the dropout operation, which consists of a 1 × 1 convolution operation and a 2 × 2 MaxPool operation. The specific network structure and parameters of the Transition layer are shown in Table 2.

Table 2 The structure and parameters of the transition layer.

Type	Parameters	
Convolution	Kernel size = 1, stride = 1	
Max pooling	Kernel size = 2, stride = 2	
Batch normalization	–	
ReLU	–	

A Dense Dilated Block is also used in the improved FCdDNet network. Specifically, dilated convolution is used on the basis of DenseNet to replace the original convolutional layer. The improvement in this part aims to replace the Layer containing dropout with a convolutional layer, and a 1 × 1 convolution is also used to integrate the final output features. The specific structure and parameters of the Dense Dilated Block used in this article are shown in Fig. 5. The Dense Dilated Block of the proposed scheme adopted a four-layer structure with dilation ratios of 2, 4, 8, and 16, respectively. Compared with the pooling layer to expand the receptive field by reducing the size of the feature map, the introduction of the dilation ratio in the Dense Dilated Block can increase the receptive field while keeping the size of the feature map unchanged, which can make the Dense Dilated Block replace the downsampling and upsampling processes in the conventional neural network, so as to avoid the loss of accuracy caused by the reduction and re-enlargement of the feature map in the process of image downsampling and upsampling. Therefore, the steganography model proposed in this article uses the dense connection and dilated convolution of Dense Dilated Block to avoid feature loss, and the improved cross-layer connection in FCdDNet further improves the feature utilization, so as to ensure the accuracy of information extraction in the decoding process.

Figure 5 Dense Dilated Block structure and parameters (L was the dilation rate).

The process of information hiding and extraction is shown in Algorithm 1. Specifically, during the information hiding stage, the secret image Ib first passes through the Transition layer, and the image size becomes 1/4 of the original size after max pooling. It is then channel-connected with the carrier image Ia. The connected data is used as the input of the improved FCdDNet in the hidden network, and through a series of convolutions, channel connections, max pooling and deconvolution in the network model, a hidden image I'a is generated. Guided by the loss function, the generated hidden image I'a will become increasingly similar to the carrier image Ia. During the information extraction stage, the hidden image I'a is used as the input of the improved FCdDNet in the extraction network, which extracts the compressed secret image. The compressed data is then decompressed through a layer of deconvolution, restoring it to the size of the original secret image Ib, resulting in the extracted image I'b. Similarly, as the model converges, the extracted image I'b will become increasingly consistent with the original secret image Ib.

Algorithm 1 The hidden and extraction process. (w, the parameters of the hidden network. θ, the parameters of the extraction network.)

Input: Ia, the carrier image with a size of 256 × 256. Ib, the secret image with a size of 512 × 512.	
Output: I’a, the hidden image with a size of 256 × 256. I’b, the extracted image with a size of 512 × 512.	
While w and θ has not converged do	
  Sample a batch of Ia from carrier image dataset.	
  Sample a batch of Ib from secret image dataset.	
  I’a = fw(Ia, Ib)	
  I’b = fθ(I’a)	
  fw←∇w(MSE(Ia, I’a) + MSE(Ib, I’b))/2	
  fθ←∇θ(MSE(Ia, I’a) + MSE(Ib, I’b))/2	
end while	

Loss function

In traditional information hiding schemes, mean square error (MSE) and peak signal-to-noise ratio (PSNR) are widely used to evaluate the effect of information hiding. MSE calculates the average square of the pixel value difference between two images, which is used to measure the difference between the two images. The minimum value of MSE is 0, which means that the two images are exactly the same. The MSE is used as the loss function in this article and does not require a complicated combination, because traditional watermarking widely uses MSE as an evaluation index. The calculation formula of MSE is as follows:

(1) MSE(I,I′)=1M×N∑i=1M⁡∑j=1N⁡(Ii,j−I′i,j)2.

Among them, I and I' are two images of length M and width N. The loss of the information hiding network is as follows:

(2) loss1=MSE(Ia,I′a).

Among them, Ia and I'a represent the carrier image and the hidden image respectively. The loss of information extraction is as follows:

(3) loss2=MSE(Ib,I′b).

Among them, Ib and Ib′ represent the secret image and the extracted image, respectively. The overall loss is obtained jointly by loss1 and loss2, and the calculation method is as follows:

(4) Loss=loss1+loss22.

Pixel overflow prevention

In the improved FCdDNet network, the output layers of the two networks used ReLU and LeakyReLU as activation functions, respectively. The results obtained by these two activation functions will exceed the pixel value range of the image. When an element is less than 0 in the result, the corresponding pixel value will also be less than 0, which is called pixel underflow. When an element is greater than 1 in the result, the corresponding pixel value will be greater than 255, which is called pixel overflow.

In order to solve the overflow problem, the method used in this article is designed to forcefully convert the obtained result so that its value range falls between the interval [0, 1]. As such, during the training process, the network will learn this change, so that the value range is as close as possible to [0, 1]. This coercion avoids pixel overflow and improves the accuracy of the network.

Experiment and results

In this article, the training set and test set used are from the VOC 2012 dataset (Everingham et al., 2015) and ImageNet dataset (Russakovsky et al., 2015). The carrier image and the secret image each have 40,000 images for training and 6,000 images for testing. The carrier image size is 256 × 256, and the secret image is expanded to 512 × 512. In this experiment, the hardware used is GPU: Tesla P100, the initial network learning rate lr = 0.001, and the number of training epochs = 500.

Example images from the dataset used in this article are shown in Fig. 6, where the upper row is the carrier image (256 × 256) and the lower row is the secret image (512 × 512).

Figure 6 Examples of test set images: carrier images (upper row) and secret images (lower row).

Image and histogram

The histogram of an image shows the number of each pixel (0–255) in the image, which can show the distribution and trend of pixels in the image. In the field of information hiding, if the histogram of the hidden image is significantly shifted from the histogram of the original carrier image, the visual effect will be poor. Figure 7 shows part of the image in the test set and its corresponding histogram.

Figure 7 Comparison of experimental images and histograms.

As shown in Fig. 7, red represents the R channel in the image, green represents the G channel in the image, and blue represents the B channel in the image. After hiding the image, the amount of modification to the original carrier image is minimal, and there is no discernible difference between the two images, at least not visible to the human eye. The extracted images are also very accurate, and there are also no discernible differences. Still in Fig. 7, whether it is hidden or extracted, the histogram distribution state of the obtained image remains almost the same, and the pixel difference is very small. At the same time, without the original carrier image (usually there is no), there is no way to tell whether a secret image is hidden or not based on the histogram.

Residual image

The residual image is an image composed of the difference between each pixel of the two images, which can reflect the difference between each pixel or the subtle position of the two images. On the other hand, if the outer contour or any residue of the secret image can be seen in the residual image, it means that the hidden network effect is poor and has no secrecy. The results of the proposed scheme are shown in Fig. 8. In order to show the residual image more clearly, the size of the secret image is zoomed out to display.

Figure 8 The residual image between the original carrier image and the hidden image, and the secret image (zoom out to display).

The residual image in Fig. 8 is basically all black and has no visual meaning, which shows that the difference between the corresponding pixels of the carrier image and the hidden image is minimal. In fact, the difference between any pixel of the two does not exceed 10. According to the 20 times residual image, it can be seen that the image contains some contour of the carrier image, but there is no relevant information about the secret image, so the setting of the hiding network can be considered optimal. More importantly, the original carrier image will not be obtained during general use, so the security of this scheme is also beyond expectation.

Hidden capacity analysis

Hidden capacity is the main standard of information hiding technology. The solution proposed in this paper has a larger capacity, and the information volume ratio of the carrier image and the secret information is 1:4. If the effective capacity (EC) (bits per pixel, bpp) is used to measure capacity, the proposed solution has an EC of 96 bpp (512 × 512 × 3 × 8 bits/256 × 256 pixel). Table 3 shows the comparison of the hidden information capacity of this scheme with the existing information hiding schemes.

Table 3 Comparison of the hidden information capacity between the proposed scheme and similar schemes.

	Schemes	Carrier image size	Secret data size	EC (bpp)	
Traditional	Abdel-Aziz, Hosny & Lashin (2021)	256 × 256 (RGB)	20 (bits)	<1	
Meng et al. (2021)	512 × 512 (gray)	<256 × 256 × 3 (bit)	<2	
DNN	Huo et al. (2024)	256 × 256 (grary)	256 × 256 × 3 × 8 (bit)	24	
Guan et al. (2022)	256 × 256 (RGB)	256 × 256 × 3 × 8 × 2 (bit)	48	
Proposed	256 × 256 (RGB)	512 × 512 × 3 × 8 (bit)	96	

According to Table 3, it can be seen that the current DNN-based schemes are much higher in capacity than traditional information hiding or watermarking schemes. However, this comparison is unreasonable, because the information hiding scheme based on DNN cannot reconstruct the secret information losslessly. The EC of this scheme is higher than other current DNN-based schemes, which is an improvement to the capacity.

But in fact, Baluja (2020) discussed that the hiding method of similar schemes is not simply to modify the image’s least significant bit (LSB), but to disperse the secret information into the image after being compressed by network coding. Therefore, the actual modified data volume of the carrier image should be analyzed, including the total amount of R, G, and B channels. After analyzing all experimental data, the average value obtained is 305,291 bits, which is the amount of data actually embedded in the carrier image.

Comparative analysis of PSNR and SSIM

The traditional information hiding field used the Peak Signal-to-Noise Ratio (PSNR) and Structural Similarity Index (SSIM) to determine the quality of hidden images. This article also uses PSNR and SSIM as criteria to evaluate the impact of hidden secret images on carrier images and to quantitatively analyze the extraction accuracy. The calculation method of PSNR is as follows:

(5) PSNR=10×log10((2n−1)2MSE(I,Ia)).

Among them, I and Ia are the two images involved in the calculation. SSIM defined and quantified the degree of structural information degradation between two images in terms of luminance, contrast, and structure. The calculation method of SSIM is as follows:

(6) SSIM(I,Ia)=(2μIμIa+c1)(2σIσIa+c2)(μI2+μIa2+c1)(σI2+σIa2+c2).

Among them, μI and μIa are the average values of I and Ia, respectively, σI and σIa are the covariances of I and Ia, and σI2 and σIa2 are the variances of I and Ia, c1 and c2 are variables that stabilize the denominator to avoid zero denominator.

When the hidden image has less influence on the carrier while the extracted image exhibits high accuracy, the value of PSNR will be relatively large and the SSIM will be close to 1. The PSNR and SSIM comparison with DNN-based information hiding schemes is shown in Table 4.

Table 4 Comparison of PSNR and SSIM between the proposed scheme and similar schemes.

Schemes		Hidden image	Extracted image	
Huo et al. (2024)
EC = 24	PSNR	33.34	32.07	
SSIM	0.93	0.93	
Guan et al. (2022)
EC = 48	PSNR	40.31	36.63	
SSIM	0.98	0.96	
Proposed
EC = 96	PSNR	42.67	36.45	
SSIM	0.98	0.95	

When the PSNR is greater than 30 dB, the image quality is better, and the difference between the images is not visible visually. As can be seen from Table 4, the PSNR values of the hidden and extracted images of the proposed scheme and the comparison method adopted in this article are greater than 30 dB, and the obtained images have good visual effects. The PSNR and SSIM values of the hidden images generated by the proposed scheme are higher than those of all comparison schemes. Therefore, it can be seen that this scheme has the best hiding effect compared to other DNN schemes. In the extraction stage, the PSNR and SSIM of the proposed scheme were higher than those of Huo et al. (2024) and slightly lower than that of Guan et al. (2022), but the proposed scheme had greater advantages in EC, more than twice that of the method of Guan et al. (2022), so the proposed scheme still had advantages overall. Therefore, compared with the existing DNN-based information hiding schemes, this scheme has certain advantages in PSNR and SSIM, and the amount of hidden information is much larger than that of similar schemes.

Effects analysis of special image

For some special images, such as the carrier image is a pure white image, we have also done related experiments, and the experimental results are shown in Fig. 9, from which it can be seen that images with more complex contours generally perform poorly, while images with fewer or more regular contours perform better.

Figure 9 Experimental results of different smoothness images (the image size of the last three columns was zoomed out for better display).

According to Fig. 9, it can be seen that the scheme has a good effect of hiding in a smooth image, even if the carrier image is completely white. Although there are obvious secret image outlines in the residual image, it is meaningless to hide the image in the pure white carrier image. This will also not be done in practical applications. However, when an image without any natural rule is used as a carrier image, the experimental effect is extremely poor. This is because the natural images in VOC 2012 and ImageNet used in training are relatively smooth and have certain natural image rules. Similar to traditional watermarking schemes, the human visual system (HSV) contends that the human eye cannot detect subtle changes in smoother areas (Meng et al., 2021). Therefore, images with relatively smooth texture and few texture outlines can have a better hiding effect, while images with cluttered textures (such as encrypted images) will have a poor hiding effect, which is also a shortcoming of this scheme.

Conclusions

In order to explore the larger hidden capacity of information hiding, an information hiding scheme based on improved FCdDNet is proposed, which can hide large-size secret images into small-size carrier images. The improved FCdDNet network serves as the shared structure of the hidden network and the extraction network. Using the method of adversarial training, the two networks influenced each other and promoted each other, so that there are very few features hidden in the hidden image, while the extracted image has high accuracy. It can be seen from the simulation experiments that even under the premise of large hidden capacity, the proposed scheme still has a certain performance compared with some existing information hiding schemes (even schemes with small capacity) in terms of evaluation criteria such as PSNR and SSIM. However, the secret images extracted by the proposed model are not accurate enough. In the future, we will attempt to preprocess and encrypt the secret images to improve the accuracy and security of the extracted images.

Additional Information and Declarations

Competing Interests

Author Contributions

Data Availability

The authors declare that they have no competing interests.

Lianshan Liu conceived and designed the experiments, performed the experiments, analyzed the data, performed the computation work, prepared figures and/or tables, authored or reviewed drafts of the article, and approved the final draft.

Li Tang conceived and designed the experiments, performed the experiments, analyzed the data, performed the computation work, prepared figures and/or tables, authored or reviewed drafts of the article, and approved the final draft.

Shanshan Tong conceived and designed the experiments, performed the experiments, analyzed the data, performed the computation work, prepared figures and/or tables, authored or reviewed drafts of the article, and approved the final draft.

Yu Huang performed the experiments, analyzed the data, performed the computation work, prepared figures and/or tables, authored or reviewed drafts of the article, and approved the final draft.

The following information was supplied regarding data availability:

The PASCAL Visual Object Classes Challenge (VOC) 2012 dataset is available at http://host.robots.ox.ac.uk/pascal/VOC/voc2012/index.html.

The ImageNet Large Scale Visual Recognition Challenge (ILSVRC) 2012 dataset is available at: https://image-net.org.

The source code is available at figshare: t, s (2023). FCdDNet-code.zip. figshare. Software. https://doi.org/10.6084/m9.figshare.24885837.v1.

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
