# Peer review of "A scheme of hiding large-size image into small-size image based on FCdDNet"

_PeerJ Computer Science, doi:10.7717/peerj-cs.2140_

## Round 0.1 · original submission · Major Revisions

Please check the reviewers' suggestions and revise the paper.

Reviewer 1 ·

Basic reporting

The paper proposed an information hiding scheme based on an improved FCdDNet. However, some problems should be addressed as follows:
1. The core optimization section of the method presented in this article is not detailed enough, and it is suggested that the description of this section be strengthened.
2. The theoretical feasibility analysis of the proposed optimal structure is missing.
3. The format of the paper should be noted, such as two-end alignment.
4. Some related references should be cited as follows:
Li H, Dong S. Image steganalysis algorithm based on deep learning and attention mechanism for computer communication[J]. Journal of Electronic Imaging, 2024, 33(1): 013015-013015.
Singh H K, Singh A K. Digital image watermarking using deep learning[J]. Multimedia Tools and Applications, 2024, 83(1): 2979-2994.

Experimental design

no comment

Validity of the findings

no comment

Cite this review as

Reviewer 2 ·

Basic reporting

The author proposes a method of hiding large-sized color images into small-sized color images. This method utilizes an improved FCdDNet network as the main structure for hiding network sharing, which can achieve 4 times the hiding of information images carrying images.

Experimental design

This manuscript lacks comparisons with the latest research results in the experimental section, and most of the comparison methods are from three years ago.

Validity of the findings

Actually, the proposed method has a certain degree of innovation, and the experiments are also detailed. However, there still are some processes need to be detailed and some mistakes need to be corrected.

Additional comments

The author did not reveal why the method proposed in this article can hide information images that are four times larger than those carrying images and extract high-quality images.

Cite this review as

Reviewer 3 ·

Basic reporting

A scheme of hiding large-size image into small-size image
based on FCdDNet
1. Research needs to make an algorithms of the improvement method
2. Which application used the algorithms of the paper?
3. research needs proofreading.

Experimental design

good

Validity of the findings

it's OK

Additional comments

keywords are necessary for the research structure

Annotated reviews are not available for download in order to protect the identity of reviewers who chose to remain anonymous.
Cite this review as

---

## Round 0.2 · Minor Revisions

Please read the reviewer's suggestions and revise the paper.

Reviewer 1 ·

Basic reporting

Although the paper has addressed my questions. However, some problems still need be addressed as follows:
1. The author should provide the core algorithm pseudocode of the method proposed in this article and provide a detailed discussion.

Experimental design

no comment

Validity of the findings

no comment

Additional comments

no comment

Cite this review as

---

## Round 0.3 · accepted · Accept

The paper has addressed all reviewer's question.

Reviewer 1 ·

Basic reporting

no comment

Experimental design

no comment

Validity of the findings

no comment

Additional comments

no comment

Cite this review as